# Costs of cannabis testing compliance: Assessing mandatory testing in the California cannabis market

**Pablo Valdes-Donoso** [1]*, **Daniel A. Sumner**[1,2], **Robin Goldstein**[1]

1 Agricultural Issues Center, University of California, Davis, California, United States of America,
2 Department of Agriculture and Resource Economics, University of California Davis, Davis, California, United States of America

* pvaldesdonoso@ucdavis.edu

**Data Availability Statement:** All relevant data are within the manuscript.

**Funding:** The author(s) received no specific funding for this work.

## Abstract

Most U.S. states that have regulated and taxed cannabis have imposed some form of mandatory safety testing requirements. In California, the country's largest and oldest legal cannabis market, mandatory testing was first enforced by state regulators in July 2018, and additional mandatory tests were introduced at the end of 2018. All cannabis must be tested and labeled as certified by a state-licensed cannabis testing laboratory before it can be legally marketed in California. Every batch that is sold by licensed retailers must be tested for more than 100 contaminants, including 66 pesticides with tolerance levels lower than the levels allowable for any other agricultural product in California. This paper estimates the costs of compliance with mandatory cannabis testing laws and regulations, using California's testing regime as a case study. We use state government data, data collected from testing laboratories, and data collected from lab equipment suppliers to run a set of Monte Carlo simulations and estimate the cost per pound of compliance with California's new cannabis testing regulations. We find that cost per pound is highly sensitive to average batch size and testing failure rates. We present results under a variety of different assumptions about batch size and failure rates. We also find that under realistic assumptions, the loss of cannabis that must be destroyed if a batch fails testing accounts for a larger share of total testing costs than does the cost of the lab tests. Using our best estimates of average batch size (8 pounds) and failure rate (4%) in the 2019 California market, we estimate testing cost at $136 per pound of dried cannabis flower, or about 10 percent of the reported average wholesale price of legal cannabis in the state. Our findings explain effects of the testing standards on the cost of supplying legal licensed cannabis, in California, other U.S. states, and foreign jurisdictions with similar testing regimes.

## Introduction

U.S. state markets for cannabis are evolving rapidly. As of mid-2019, 32 of 50 states had some form of legal medicinal cannabis system in place, and since 2012, 11 of those states had legalized and regulated adult-use cannabis [1].

**Competing interests:** The authors have declared that no competing interests exist.

California was the first U.S. state to decriminalize the sale of medicinal cannabis, with the 1996 passage of the Compassionate Use Act (Proposition 215). In 2003, a California state legislative act, Senate Bill 420, set out more specific rules for the operation of medicinal cannabis collectives and cooperatives. For the following 15 years, regulations on the cultivation, manufacturing, and sale of cannabis in California were largely limited to a wide variety of local ordinances, with little intervention from the state government.

In November 2016, California voters legalized adult-use cannabis by approving Proposition 64 (the Adult Use of Marijuana Act, or AUMA). Subsequently, the Medicinal and Adult-Use Cannabis Regulation and Safety Act of 2017 (MAUCRSA) created a unified framework for the state licensing of cannabis businesses and the taxation and regulation of adult-use and medicinal cannabis. MAUCRSA regulations went into effect on January 1, 2018 [2].

Safety regulations generally add costs to production. One of the most costly components of California's new system of cannabis regulation is the mandatory testing of all legal cannabis for more than 100 contaminants, including pesticides and heavy metals. This paper is the first to comprehensively examine the economic challenges of cannabis testing and estimate the cost of testing compliance per pound of cannabis marketed in a legal and licensed cannabis market. In a previous article [2], we provide a brief introduction to testing costs to which this paper supplies needed rigor.

We review and compare the allowable tolerance levels for contaminants in cannabis with allowable levels in other crops from California, and review rejection rates in California since mandatory testing began in 2018. We compare these with rejection rates in other U.S. states where medical and recreational use of cannabis are permitted. We use primary data from California's major cannabis testing laboratories, several cannabis testing equipment manufacturers, Bureau of Cannabis Control license data including geographical location information, and data from Cannabis Benchmarks on average wholesale batch sizes to estimate the testing cost per pound of cannabis legally marketed in California.

## Background

At the U.S. federal level, cannabis is still classified as a Schedule I illegal narcotic, and its possession, sale, and even testing are serious criminal offenses under federal law [3,4]. Even cannabis businesses that are fully compliant with state regulations thus face legal risks, uncertainties, and obstacles to doing business such as a lack of access to mainstream banks [5]. In recent years, however, the conflict of state and federal laws has generally been mediated via a series of informal, non-binding agreements, letters, and memos of understanding between the U.S. Department of Justice and states. These understandings have enabled cannabis businesses to focus more on complying with state and local laws than on hiding from federal prosecutors.

All of the U.S. states that have legalized, taxed, and regulated recreational cannabis, and most states that have legalized and regulated medicinal cannabis, require testing for some contaminants (e.g. fungus, mold, bacteria, and mycotoxins) and testing and labeling of potency (as measured by concentrations of cannabinoids such as the phytocannabinoid tetrahydrocannabinol, or THC, and cannabidiol, or CBD). Colorado and Washington were the first states to vote to legalize and regulate adult-use cannabis, both in 2012. Colorado first introduced the enforcement of potency and homogeneity tests for retail cannabis products in 2014. Residual solvents and microbial contaminants were added to the testing requirements in 2015, and heavy metals and pesticide residues as of mid-2018 [6–8]. Washington State mandates that licensed testing laboratories must also perform potency tests, moisture analysis, foreign matter,

microbial and mycotoxin screenings, and (for extracted cannabis, e.g. cannabis oil) screenings for residual solvents [9].

Some states, including California and Colorado but not Washington, also require more sophisticated and costly wet-lab tests for pesticides and heavy metals. Per MAUCRSA, the California Department of Pesticide Regulation (DPR) established maximum allowable thresholds for 66 different pesticides, including zero tolerance for trace amounts of 21 pesticides and low allowable trace amounts of 45 other pesticides. MAUCRSA also established thresholds for 22 residual solvents plus a variety of heavy metals and other contaminants. The Bureau of Cannabis Control (BCC) was put in charge of licensing and regulating testing labs and enforcing the testing standards.

In the 2016 marketplace, prior to the passage of Proposition 64—which was unregulated at the state level and partially regulated (medicinally) at the local level—total California cannabis production was estimated at approximately 13.5 million pounds of raw flower, with roughly 80% of this production illegally shipped out of the state [10]. These out-of-state shipments may explain why California accounted for 70% of nationwide cannabis confiscations in 2016 [11]. Rough estimates suggest that only about one-quarter of California's in-state cannabis consumption, or less than 5% of total cannabis production, went to the legal medicinal market in 2016 [12].

Until 2018, there were no rules in place at the state or local levels in California for testing contaminants, even for products legally marketed as medicinal cannabis [5,13]. A minority of medicinal cannabis retailers in the pre-2018 state-unregulated market was routinely testing and labeling cannabis for THC potency, but few were voluntarily testing for contaminants. Informal evidence suggests that pesticide residues were common in cannabis products in the pre-regulated market. For example, in 2017 an investigation reported that 93% of 44 samples collected from 15 cannabis retailers in California had pesticide residues [14].

## California's cannabis testing standards

The mandatory testing framework introduced under MAUCRSA is summarized in Table 1, where we briefly describe the tests for specific types of batches and the standards for passing each test. Dried cannabis flower (or "harvest batches") and cannabis products (or "manufactured batches") must be tested for concentrations of cannabinoids and various contaminants in order to enter the legal market. Some tests (for cannabinoids, presence of foreign material, pesticides, heavy metals, mycotoxins, microbial impurities and terpenoids) apply to all batches, while some others (moisture content, residual solvents, processing chemicals) only apply to some forms of cannabis. Heavy metals tests (for traces of lead, cadmium, arsenic, and mercury) were not mandatory until December 2018.

Table 2 shows the list of contaminants with their maximum tolerance levels allowed in California. Tolerance levels are generally lower for products that are inhaled than for products that are eaten or applied topically. For 21 pesticides, the maximum residual level is zero, meaning that no trace of those residues may legally be detected in a sample of cannabis.

MAUCRSA requires that all batches of cannabis flowers and products must be sampled and tested by licensed laboratories before being delivered to retailers. Distributors are responsible for testing (although cultivators, manufacturers, and retailers may also hold distributor licenses). Fig 1 shows the flow of cannabis testing in California. The weight of a harvest batch cannot exceed 50 pounds; larger batches must be broken down into 50-pound sub-batches for testing. The sample size must be bigger than 0.35% of its weight. A processed batch cannot surpass 150,000 units.

**Table 1. Summary of mandatory testing per batch type and criterion used to pass tests.**

| Type of Test | Description | Batch Tested | Criterion Required to Pass Test |
|---|---|---|---|
| Cannabinoids[1] | Measure concentration of THC, THCA, CBD, CBDA, CBG, and CBN | All | ≤10mg of THC per serving of edibles. ≤100mg of THC per package of medical use only products. ≤500mg of THC per package of medical use only orally dissolving products. ≤1000mg of THC per package of topical products. ≤2000mg of THC per package of medical use only topical products. |
| Foreign material[2] | Determine presence of foreign material (hair, insects, feces, packaging contaminants, and manufacturing waste) | All | ≤ 1/4 of sample area covered by sand, soil, cinders, dirt, mold, or any imbedded foreign material. ≤ 1 insect fragment, hair, or count mammalian excreta per 3g |
| Pesticide[3] | Absence of 21 and limited presence of 45 pesticide residues | All | Levels of specific contaminants below action levels (see Table 2) |
| Heavy metals[4] | Limited presence of four heavy metals | All | Levels of specific contaminants below action levels (see Table 2) |
| Mycotoxins[4] | Screening of Aflatoxin B1, B2, G1, and G2, and Ochratoxin A | All | Aflatoxin B1, B2, G1, and G2, and Ochratoxin A < 20 μg/g |
| Microbial impurities[1] | Screening of Shiga toxin -*Escherichia coli*, *Salmonella* spp., and pathogenic *Aspergillus* species | All [†] | Shiga toxin and *Salmonella* spp., and *Aspergillus* species (*A. fumigatus*, *flavus*, *niger*, and *terreus*), undetected in 1g |
| Moisture content[1] and water activity[4] | Measure of moisture content, and water activity ($A_w$) according with type of product | Flowers, solid edible products | $A_w \leq 0.65$ for dried flowers or ≤ 0.85 for solid and semi-solid edible products. Lab must report moisture content as a percentage. |
| Solvent and processing chemical[3] | Absence of six and limited presence of 14 solvent and processing chemical residues | Cannabis products or pre-rolls | Levels of specific contaminants below action levels (see Table 2) |
| Terpenoids[4] | Determination if sample conforms to the labeled content of terpenoids | All labeled products | The laboratory must report the result of the terpenoid testing as a percentage in mg/g or mg/mL depending on the type of product. |

**Source:** MAUCRSA and California regulations (November 2018)

[†] Screening of *Aspergillus* species only in inhalable cannabis or inhalable cannabis products.

Phase-In of Testing Regime:

[1] All cannabis harvested or manufactured on or after January 1, 2018, must have been tested for this item.

[2] In addition to the analytical tests required in [1], all cannabis harvested or manufactured on or after July 1, 2018, must have been tested for this item.

[3] Category I of pesticides and II of and solvents-and-processing chemicals must have been tested for cannabis harvested or manufactured on or after January 1, 2018. Category II of pesticides and I of solvents-and-processing chemicals must have been tested for cannabis harvested or manufactured on or after July 1, 2018.

[4] In addition to the analytical tests required in [1], [3], and [3], all cannabis harvested or manufactured on or after December 31, 2018, must have been tested for this item.

After testing each batch, laboratories must file a certificate of analysis (COA) indicating the results to distributors and to the BCC. If a sample fails any test, the batch that it represents cannot be delivered to dispensaries for marketing. Instead, it can be remediated or reprocessed (no more than twice) and fully re-tested again. If a batch fails a second re-testing after a second remediation, or if a failed batch is not remediated, then the entire batch must be destroyed.

Analyzing the cannabis market, compared with other agricultural markets, presents a unique challenge to researchers because of the rapidly changing legal environment, the lack of historical data or scientific studies, the lack of government tax data, and the cash nature of the business. Testing prices are not publicly advertised by licensed laboratories. Quotes are known to vary depending on the number of samples, the frequency of testing, the type of contract between the distributor and the laboratory, among others. Bulk pricing is common and is negotiated on a case-by-case basis.

We approximate the costs of testing by collecting detailed data on the testing process and constructing in-depth estimates of the capital, fixed, and variable costs of running a licensed testing laboratory in California. We use these results in a set of simulations that estimate the costs per pound generated by cannabis testing under the California regulations in place as of mid-2019. We make some market assumptions based on the most reliable industry data available as of this writing in order to estimate the current cost per pound of testing compliance.

**Table 2. Tolerance levels for pesticide residues, heavy metals, residual solvents, and processing chemicals in cannabis and cannabis products in California.**

| Type of Contaminant | Tolerance Level (μg/g)[2] |
|---|---|
| *Pesticide residues* | |
| Aldicarb, Carbofuran, Chlordane, Chlorfenapyr, Chlorpyrifos, Coumaphos, Daminozide, DDVP (Dichlorvos), Dimethoate, Ethoprop(hos), Etofenprox, Fenoxycarb, Fipronil, Imazalil, Methiocarb, Methyl parathion, Mevinphos, Paclobutrazol, Propoxur, Spiroxamine, Thiacloprid[1] | 0 (I), 0 (O) |
| Acephate, Acetamiprid, Bifenazate | 0.1 (I), 5 (O) |
| Abamectin | 0.1 (I), 0.3 (O) |
| Acequinocyl | 0.1 (I), 4 (O) |
| Azoxystrobin | 0.1 (I), 40 (O) |
| Bifenthrin | 3 (I), 0.5 (O) |
| Boscalid | 0.1 (I), 10 (O) |
| Captan | 0.7 (I), 5 (O) |
| Carbaryl | 0.5 (I), 0.5 (O) |
| Chlorantraniliprole | 10 (I), 40 (O) |
| Clofentezine | 0.1 (I), 0.5 (O) |
| Cyfluthrin | 2 (I), 1 (O) |
| Cypermethrin | 1 (I), 1 (O) |
| Diazinon | 0.1 (I), 0.2 (O) |
| Dimethomorph | 2 (I), 20 (O) |
| Etoxazole | 0.1 (I), 1.5 (O) |
| Fenhexamid | 0.1 (I), 10 (O) |
| Fenpyroximate, Flonicamid, Hexythiazox | 0.1 (I), 2 (O) |
| Fludioxonil | 0.1 (I), 30 (O) |
| Imidacloprid | 5 (I), 3 (O) |
| Kresoxim-methyl | 0.1 (I), 1 (O) |
| Malathion | 0.5 (I), 5 (O) |
| Metalaxyl | 2 (I), 15 (O) |
| Methomyl | 1 (I), 0.1 (O) |
| Myclobutanil | 0.1 (I), 9 (O) |
| Naled | 0.1 (I), 0.5 (O) |
| Oxamyl | 0.5 (I), 0.2 (O) |
| Pentachloronitrobenzene | 0.1 (I), 0.2 (O) |
| Permethrin | 0.5 (I), 20 (O) |
| Phosmet | 0.1 (I), 0.2 (O) |
| Piperonylbutoxide | 3 (I), 8 (O) |
| Prallethrin | 0.1 (I), 0.4 (O) |
| Propiconazole | 0.1 (I), 20 (O) |
| Pyrethrins | 0.5 (I), 1 (O) |
| Pyridaben, Spinetoram, Spinosad | 0.1 (I), 3 (O) |
| Spiromesifen | 0.1 (I), 12 (O) |
| Spirotetramat | 0.1 (I), 13 (O) |
| Tebuconazole | 0.1 (I), 2 (O) |
| Thiamethoxam | 5 (I), 4.5 (O) |
| Trifloxystrobin | 0.1 (I), 30 (O) |
| *Heavy metals (not mandatory until 12/31/2018)* | |
| Cadmium | 0.2 (I), 0.5 (O) |

*(Continued)*

**Table 2.** (Continued)

| Type of Contaminant | Tolerance Level (µg/g)[2] |
|---|---|
| Lead | 0.5 (I), 0.5 (O) |
| Arsenic | 0.2 (I), 1.5 (O) |
| Mercury | 0.1 (I), 3 (O) |
| *Residual solvent and processing chemicals* | |
| 1,2-Dichloroethane, Benzene, Chloroform, Ethylene oxide, Methylene chloride, Trichloroethylene[1] | 1 (I or O) |
| Acetone | 5000 (I or O) |
| Acetonitrile | 410 (I or O) |
| Butane, Ethanol, Ethyl acetate, Ethyl ether, Heptane | 5000 (I or O) |
| Hexane | 290 (I or O) |
| Isopropyl alcohol | 5000 (I or O) |
| Methanol | 3000 (I or O) |
| Pentane, Propane | 5000 (I or O) |
| Toluene | 890 (I or O) |
| Total xylenes (ortho-, meta-, para-) | 2170 (I or O) |

**Source:** MAUCRSA and California regulations (November, 2018)

[1] These pesticides or residual solvents are categorized as I, while the rest of pesticides or residual solvents are categorized as II.

[2] I denotes the tolerance levels for inhalable cannabis, and O the tolerance levels for any other type of cannabis.

## Materials and methods

We construct a simulation model using R software [15] to assess the cost structure of cannabis testing in California under the current regulatory framework. We base our simulations on the number of testing labs ($n = 49$) and distributors ($n = 1,210$) that had been granted temporary licenses by the BCC as of April 2019. The number of labs and distributors in California will fluctuate as the industry continues to develop.

To estimate costs incurred by labs, we first construct estimates of fixed and variable costs for labs based on their testing capacities. We calculate the cost of testing a sample of dried cannabis flower considering the lab scale and the distances between labs and distributors. Based on meetings with representatives of California testing labs, we assume that 70%, 20%, and 10% of the labs are distributed into small, medium, and large size categories. We assume that the testing industry is like many others in that many small firms (that have relatively high costs) supply relatively little of the output. We run 1,000 simulations to estimate the cost of sampling and testing for a sample of a typical batch of dried flowers from each of the 49 labs, assuming that costs, working hours, testing capacities, etc., may vary from lab to lab.

Next, we use the weighted average of testing cost per sample to estimate cost per pound. We express total testing cost in dollars per pound of legal cannabis that reaches the market, after incorporating costs of remediating and re-testing failed batches and losses from batches of cannabis that cannot be remediated and must be destroyed.

## Data

We used the list published by the BCC [16] to identify actively licensed testing labs and requested, to managers or representatives, a personal or phone-call interview based on a set of questions that we used as a guideline (S1 Appendix). We interviewed one-fourth of the

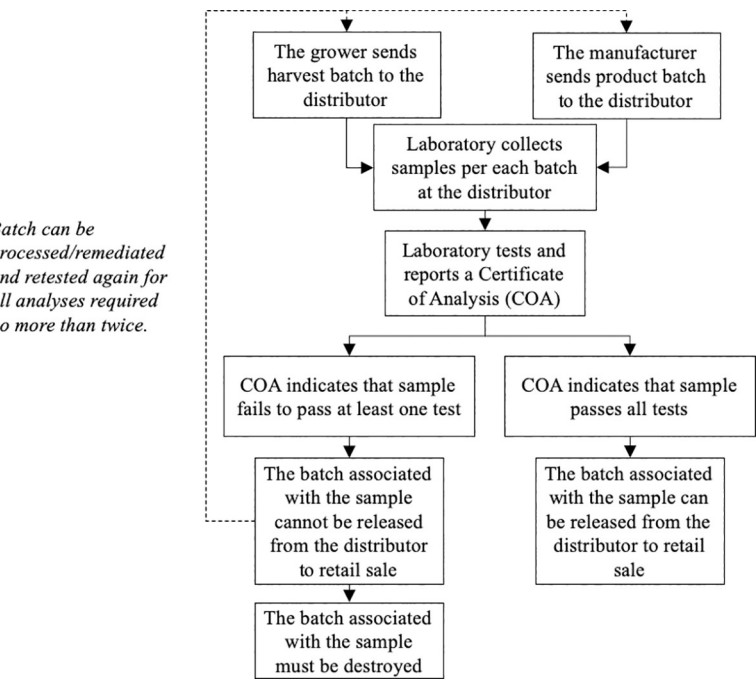

**Fig 1. California's cannabis testing process.**

operating or prospective licensed testing labs listed by the BCC. We gathered data on market prices for testing equipment, supplies and chemical reagents consumed by equipment, equipment running capacities, and other cannabis testing inputs needed to build a compliant testing laboratory in California. Likewise, we collected financial, managerial, and logistics data. To complement licensed testing lab data, we also drew on personal interviews, phone calls, and email exchanges with sales representatives of three large equipment suppliers. Table 3 summarizes capital costs, other one-time expenses, and annual operational and maintenance costs used in our calculations. We report average cost and standard deviation for each estimate.

We assume that medium-sized and large labs receive discounted prices on equipment, given the larger scale of their purchases. Based on information provided by equipment suppliers, we expect these discounts to be between 1.5% and 2.5%. Different-sized labs have different capacities based on their scale. We assume that larger labs have made larger capital investments and are better able to optimize processes when supplying a larger volume of testing (for instance, by minimizing lab downtime through redundancy in equipment). On the other hand, small testing labs require less equipment and less capital investment, and operate with low annual costs, but their testing capacities are also low. Table 4 summarizes our estimates of running time for tests, the main consumables used by testing machines, and the expected cost of running a specific test per sample. In addition, we include a range of $80 to $120 per sample to cover general material and labor apparel (e.g., tubes, glassware, goggles, masks, gloves, etc.) used while preparing and processing samples.

## Calculating laboratory cost per sample of cannabis

We break down the laboratory cost per successfully tested sample ($C_i$) of each lab ($i = 1, \ldots, 49$) into two major components: the per-sample cost of testing itself ($A_i$), and the per-sample cost of collecting, handling and transporting samples from distributors to labs ($S_i$), thus $C_i = A_i + S_i$.

**Table 3. Average expected costs of implementing a licensed cannabis testing laboratory in California.**

| Item | Range[1] | Cost[2] |
|---|---|---|
| *Cannabis testing equipment costs* | | |
| High-performance liquid chromatography (HPLC) | 1–2 | $85,195 ($20,261) |
| Inductively coupled plasma mass spectrometry (ICP-MS) | 1–2 | $259,874 ($11,847) |
| Liquid chromatography-mass spectrometry (LCMS) | 1–3 | $502,807 ($50,255) |
| Gas chromatography-mass spectrometry (GCMS) | 1–4 | $135,258 ($8,533) |
| Real-time polymerase chain reaction test (PCR) | 1–2 | $50,071 ($4,995) |
| Moisture balance (MB) | 1–2 | $9,994 ($1,019) |
| *Other one-time costs* | | |
| Start-up and support equipment | 2–4 | $237,284 ($50,176) |
| Sampling equipment and vehicles | 1–5 | $19,927 ($4,011) |
| ISO implementation IQ&OQ protocols | 1–2 | $249,257 ($24,356) |
| *Annual costs of operations and maintenance* | | |
| Compliance with ISO/IEC-17025 standards | | $24,990 ($577) |
| Maintenance[3] | | $347,785 ($150,166) |
| Real estate | | $317,549 ($124,284) |
| Utilities | | $72,535 ($29,579) |
| Sales, general & administrative costs | | $89,335 ($32,847) |
| Lab fees | | $51,667 ($28,968) |
| *Annual salary and benefits* | | |
| Director | 1–1 | $133,274 ($8,973) |
| Manager | 0–3 | $94,867 ($6,963) |
| Analyst | 2–5 | $65,011 ($6,470) |
| Technician | 2–5 | $45,654 ($3,784) |
| Sampler | 1–5 | $45,993 ($4,038) |
| Staff | 2–4 | $44,000($3,566) |

*Source*: Equipment suppliers and licensed testing laboratories.

[1] Range (minimum and maximum) of equipment and labor used depending on the size of each lab (small, medium, or large).

[2] Values represent the average cost, with standard deviations in parentheses.

[3] Maintenance cost depends on the number of equipment of each lab.

First, we consider the cost of testing per sample, which is equal to the annual total cost of running a testing lab ($AC_i$), plus the associated total return to risk and profit for the lab, divided by $Q_i$, the number of samples successfully tested per year. $Q_i$ is defined and explained below. The return to risk and profit is calculated as rate of return $\pi$ (e.g. 15%), multiplied by the amount of invested capital ($F_i$).

$$A_i = \frac{AC_i + F_i\pi}{Q_i} \tag{1}$$

Notice that invested capital enters per-unit costs in two places: one is a separate term for the rate of return $\pi$ times the amount of invested $F_i$, and the other is a component of the cost of operating the lab.

The annual cost of running a testing lab ($AC_i$) includes the annualized cost of capital invested ($F_i$) (equipment and other one-time expenses, including startup and sampling equipment, and implementation of ISO), annual operational and maintenance costs ($OM_i$), and annual labor costs ($L_i$), thus $AC_i = F_i + OM_i + L_i$ (see Tables 3 and 4 for detailed costs).

Table 4. Main consumables used by each cannabis testing lab machine, estimated testing time and cost per sample.

| Machine | Type of Analysis | Testing Time | Consumables | Cost per test[1] |
|---|---|---|---|---|
| High-performance liquid chromatography (HPLC) | Cannabinoids | 10 min | Water grade, acetonitrile, chemicals used for extraction and cleanup | $1.57 ($0.34) |
| Inductively coupled plasma mass spectrometry (ICPMS) | Heavy metals | 5 min | Argon gas, chemicals used for extraction and cleanup | $1.52 ($0.29) |
| Liquid chromatography-mass spectrometry (LCMS) | Residual pesticides and mycotoxins | 12 min | Liquid Nitrogen, water grade, methanol, chemicals used for extraction and cleanup | $8.82 ($1.05) |
| Gas chromatography-mass spectrometry (GCMS) | Residual pesticides and solvents | 20 min | Helium gas, chemicals used for extraction and cleanup | $2.52 ($0.31) |
| Real-time polymerase chain reaction test (PCR) | Microbial | 4 min | Chemicals used for extraction and cleanup | $7.49 ($0.87) |
| Moisture balance (MB) | Moisture | 10 min | Chemicals used for extraction and cleanup | $1.25 ($0.15) |

Source: Equipment supplier companies and licensed testing laboratories.

[1] Values represent the average cost, with standard deviations in parentheses.

We calculate the annualized flow of capital invested ($F_i$) based on a formula typically used in budgeting studies (Eq 2):

$$F_i = \frac{rK}{1 - (1 + r)^{-h}},$$

(2)

where $K_i$ denotes the total cost of capital and one-time expenses, $r$ denotes an assumed annual discount rate that reflects the combined effects of depreciation and interest (we use $r = 7.5\%$), and $h$ denotes the number of years to the investment horizon (we use $h = 10$ years).

The number of samples successfully tested per year ($Q_i$), $OM_i$, and $L_i$ all depend on the size of each lab, e.g. the number of pieces of each type of testing equipment, number of employees, working hours per employee, and so on. We assume that small-scale labs operate with the minimum necessary investment in testing equipment, maintenance, staff, and operating hours, while large-scale labs operate with much higher investment in testing equipment, maintenance, staff, and operating hours.

To estimate $Q_i$ for each lab, we make four sets of assumptions about each lab's operational efficiency. Our efficiency assumptions depend on the lab's size. First, we estimate a lab's potential single-day testing capacity $q_i$, which is the maximum number of samples per day that could theoretically be tested (successfully or unsuccessfully) by a lab's machinery if it were operating continuously. We use the testing time per sample ($TS_k$) of each piece of equipment (see Table 4) and the total testing operational hours per day of each lab ($TT_i$) to calculate $q_i$. Because large-scale labs hire and manage more workers, we assume that they conduct testing for 14 hours per day (with a standard deviation of 0.06 hours); while small and medium-scale labs conduct testing for about 8 hours per day (with a standard deviation of 0.06 hours).

We assume that multiple tests can be run from a single sample, some machines can run simultaneously, and some machines run more than one type of test (see Table 4). Based on these assumptions, we calculate the total number of samples tested per day per one (or more) machines of the same type ($nk$), such that $q_i^{nk} = \frac{nk}{TS_k} TT_i$, where $n = 1,\ldots, n$ (see Table 3), and $k = 1, \ldots, 6$ (see Table 4). We then constrain the total number of samples tested per day by using the minimum number of samples tested per one (or more) machines of the same type, such that $q_i = \min\{q_i^{n1}, \ldots, q_i^{n6}\}$.

Second, we assume that during each operating day, a lab runs at some percent operational efficiency $R_i$ of its maximum daily testing capacity $q_i$. Efficiency percent $R_i$ is defined as the ratio of actual tests per day performed by a lab to total possible tests per day $q_i$; thus $(q_i * R_i)$ gives the number of actual tests per day performed by a lab. We assume that the operational efficiency ratio $R_i$ is a function of equipment redundancy, maintenance-related downtime, efficient scheduling with customers, and other operational factors, which together are proportional to lab size. After many interviews with lab operators, we assume that $R_i$ is about 55% (with a standard deviation of 2.9%) for small labs, 70% (with a standard deviation of 2.9%) for medium labs, and 80% (with a standard deviation of 2.9%) for large-scale labs.

Third, we assume that labs, even when fully operational, will perform some unsuccessful tests (e.g., where machines fail, errors are made by lab technicians, or results are ambiguous, such that the lab is unable to issue a certificate of analysis without re-testing the sample at its own expense). We assume that small and medium-scale labs, because they have less experience and are likely to have less well-trained staff, have higher rates of unsuccessful tests than large-scale labs. We denote the ratio of successful tests to all tests as $E_i$, and we assume that this successful testing ratio $E_i$ is about 80% (with a standard deviation of 1.7%) in small- and medium-scale labs, and about 96% (with a standard deviation of 1%) in large-scale labs.

Fourth, we assume that small-scale labs operate fewer days per year than large-scale labs. In Eq (3), we denote the lab's operational days per year as $D_i$. Based on information provided by representatives of testing labs, we assume that $D_i$ is 240–260 days for small-scale labs, 250–270 days for medium-scale labs, and 280–300 days for large-scale labs.

The total number of samples successfully tested per year ($Q_i$) is expressed in Eq (3) as a function of the four sets of operational assumptions described above:

$$Q_i = q_i * R_i * E_i * D_i \qquad (3)$$

Next, we must estimate sampling cost ($S_i$), which includes transportation, labor, equipment, and material costs. We use the zip codes of active licensed testing labs (denoted by $i$) and distributors (denoted by $j$) published by the BCC [16] to estimate the distance from labs to distributors ($d_{ij}$). Using the R software [15] and the zip codes of active licensed testing labs and distributors [16], we generate a map that shows the geographical location of licensed labs and distributors in mid-2019 (Fig 2). We have no information about which labs served which distributors. However, we expect that labs are better able to compete for nearby distributors because they would have lower transportation time and cost and may be more likely to have closer business relationships.

In order to estimate average transport costs from distributors to labs, we randomly assigned distributors located within a 160 mi radius to each lab. Based on 2019 data, this was the longest travel distance from a distributor to the nearest lab. This travel distance radius ensures that each distributor in the sample is covered by at least one laboratory.

Based on the annual number of samples that we estimate each lab is able to test ($Q_i$), we estimate the share of total testing done by small labs, medium labs, and large labs. We then estimate the number of distributors per lab. In each of our 1,000 simulations, 70% of the 49 licensees (34 labs) with specific locations were randomly chosen to represent small-scale labs, 20% (10 labs) were randomly chosen to represent medium-scale labs, and 10% (5 labs) were randomly chosen to represent large-scale labs.

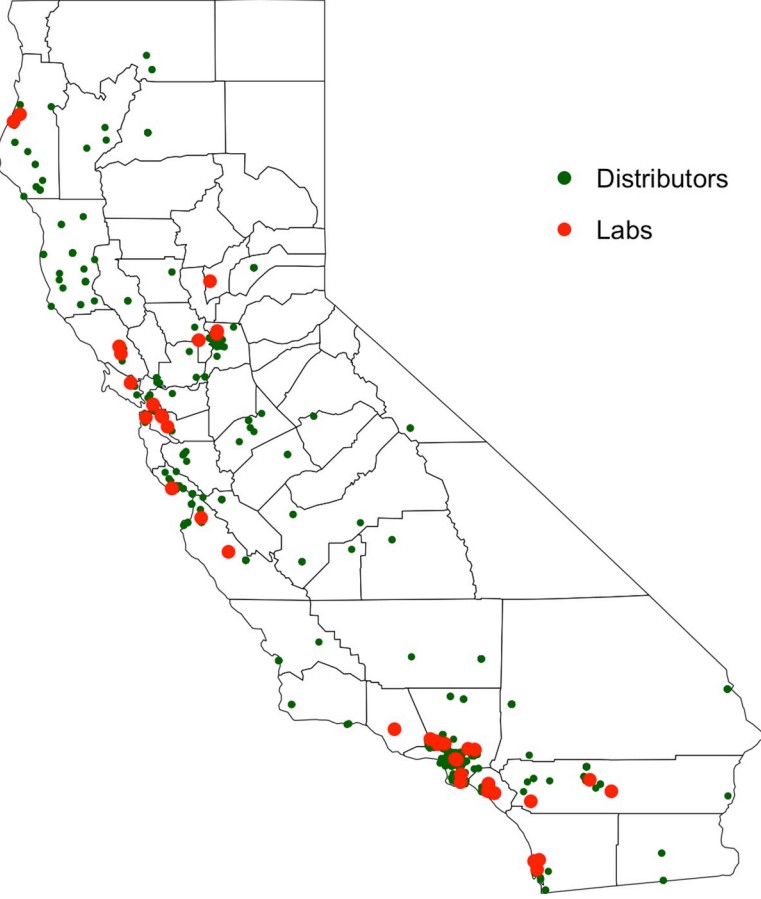

**Fig 2. Locations of licensed testing labs and distributors in California.** (Source: Bureau of Cannabis Control, April of 2019).

Eq 4 shows our calculation of sampling cost as the average cost of sampling from each distributor supplied by sampling and testing from an $i^{th}$ lab.

$$S_i = \frac{1}{n} \left( \sum_{j=1}^{n} \frac{0.535 * d_{ij} * TD_{ij}}{Q_{ij}} + \sum_{j=1}^{n} 4.5 + \frac{\frac{d_{ij}}{sp} * L_s * TD_{ij}}{Q_{ij}} + \sum_{j=1}^{n} \frac{50 * TD_{ij}}{Q_{ij}} \right) \tag{4}$$

The first term of the right hand side of Eq 4 shows transportation cost. The term $d_{ij}$ denotes the distance, back and forth, from an $i^{th}$ lab to a $j^{th}$ distributor, and $TD_{ij}$ the number of trips, per-sampler, needed to collect the total samples ($Q_{ij}$) from a particular distributor within a year. We use the cost of transportation as $0.58 per mile, the Federal reimbursement rate per mile for California. The second term of the right hand side of Eq 4 shows the labor cost of sampling. We include the time spent sampling and traveling back and forth to distributors. We use $18 per hour as the labor opportunity cost of sampling ($L_s$), and used 15 minutes as the average time used to retrieve one sample from a batch. We expect that a sampling employee will collect at least 10 samples from a distributor in one visit (i.e., $\frac{0.25hr * Q_{ij}}{Q_{ij}} * 18/hr = 4.5$). We used a range of speed ($sp$) between 50 and 60 miles per hour to calculate the time spent traveling (i.e., $\frac{d_{ij}}{sp}$). The third term of the right-hand side of Eq 4 shows the material cost of sampling. Based on

information provided by testing labs, we assume a $50 cost of materials per visit, including disposables such as gloves, masks, etc.

## Calculating cost per pound of cannabis marketed

Our estimates of cost per pound of cannabis that reaches the market ($m$) includes laboratory cost ($c$), the value of lost inventory ($v$), and the cost of remediation of failed batches ($z$): $m = c + v + z$. Laboratory cost per pound of marketed cannabis ($c$) is expressed by Eq 5:

$$c = \left(\frac{\bar{C}}{B} + \bar{G}\right) * TP,$$ (5)

where $\bar{C}$ is a weighted average of the lab cost per sample using the share of annual testing capabilities of each lab, $B$ is the batch size or number of pounds of a batch of cannabis flower from which a sample is taken for analysis, $\bar{G}$ is the weighted average of per-pound cost of security compliance, and $TP$ is the ratio of the number of pounds tested per pound that passes testing.

We use a range of values less than or equal to 50 pounds for $B$, as MAUCRSA sets a maximum batch size of 50 pounds of cannabis flower (or 150,000 units for processed cannabis products) [1]. We use a weighted average of security compliance cost of $3.92 per pound ($4.88 for $G$ in small labs, $4.06 in medium labs, and $3.25 in large-scale labs). These costs include video surveillance and archival, disposal and quarantine, and other compliance estimated elsewhere [10]. Eq 6 shows the calculation of $TP$:

$$TP = \frac{1 + pre.test\% + fail\% * re.test\%}{1 - (fail\% - fail\% * re.test\% * fail.re.test\%)}$$ (6)

The numerator of Eq 6 expresses total pounds tested (including pre-tested and rejected pounds), and the denominator expresses total pounds that pass testing. Based on our best estimates after interviewing testing labs and gathering data from other states, we set the pre-test share at 25%, the re-tested share of failed samples at 50%, and the failure rate of re-tested samples at 50%.

We use a range of failure rates from 0% to 8% based on the ranges of cannabis testing failure rates that have been observed in California in 2018 and 2019 [17], and by comparing cannabis tolerance levels with data on pesticide residue detection in other crops of California [18]. Cannabis and cannabis products are offered to be consumed via edibles or inhalable. Here we treat cannabis like any other crop in California.

Table 5 shows the percent of California food products with any detection of pesticide residues and above the U.S. Environmental Protection Agency (EPA) tolerance levels, from 2015 to 2017. We estimate that about 13% of over 7,000 samples would have been above the inhalable cannabis product tolerance limits, whereas about 4% would have exceeded even the less stringent tolerance levels established for other (non-inhalable) cannabis products (Table 5).

The value of lost inventory ($v$) includes the cost of the cannabis used up in the testing procedures ($\bar{p} * s$) and the cost of cannabis that must be destroyed when it has failed testing twice ($\bar{p} * d$). We use a wholesale price of cannabis flower of $1,360 per pound as a weighted average ($\bar{p}$) from outdoor, greenhouse, and indoor grow types [10]. Eq 7 expresses the value lost inventory per pound of cannabis that passes testing.

$$v = \frac{\bar{p}(s + d)}{1 - (fail\% - fail\% * re.test\% * fail.re.test\%)}$$ (7)

To calculate the amount of cannabis that must be removed from inventory for testing ($s$), we include a fraction (0.35%) of the batch size ($B$) that is removed for testing, and a fraction

**Table 5. Percent of California food product samples indicating any detection of pesticide residues, above EPA tolerance levels, and percent above tolerance levels for cannabis products (2015–2017).**

| Food product | 2015 | 2016 | 2017 | Total |
|---|---|---|---|---|
| With any detection of pesticide residues* | 60.35% | 60.06% | 61.46% | 60.52% |
| With pesticide residues above EPA tolerance levels* | 0.32% | 1.51% | 0.45% | 0.79% |
| **Food that would have exceeded cannabis tolerance levels** | | | | |
| Using criteria for inhalable cannabis products | 12.86% | 13.44% | 12.79% | 13.05% |
| Using criteria for other cannabis products | 4.07% | 3.62% | 3.90% | 3.86% |

*Source*: California Department of Pesticide Regulation: Pesticide Residue Monitoring Program, and MAUCRSA and California regulations, 2018.

that is retested after failing the first round of testing, and a fraction that is pre-tested (Eq 8).

$$s = (0.0035B + 0.0035B * fail\% * re.test\% + 0.0035B * pre.test\%)/B \tag{8}$$

We calculate the amount of cannabis that is destroyed after remediation (*d*) by using a fraction of cannabis that fails (between 0% and 8%) minus the share of cannabis that passes the second test (Eq 9).

$$d = fail\% - fail\% * re.test\% * fail.re.test\% \tag{9}$$

As for the value of lost inventory, Eq 10 shows how we estimate the remediation cost of failed batches at per pound of cannabis that passes testing (*z*). Based on our interviews with operators, we assume that the cost of remediation *P* (processing batches that did not pass the first round of testing) is $20 per pound.

$$z = \frac{P * fail\% * fail.re.test\%}{1 - (fail\% - fail\% * re.test\% * fail.re.test\%)} \tag{10}$$

## Results of simulations

### Laboratory cost per sample of cannabis

The minimum capital investment in testing equipment needed to satisfy regulations is substantial. We estimate that in small labs (*n* = 34), capital investment in equipment is about $1.1 million; in the medium-sized labs (n = 10), capital investment in equipment is about $1.8 million; and in large-scale labs (*n* = 5), capital investment in equipment is about $2.8 million. These capital costs, amortized over a 10-year time span with a 7.5% rate of depreciation and interest, represent less than 15% of total annual expenses. Annual costs of operating range from $1.4 to $2.2 million for small labs, $2.7 to $3.7 million for medium-sized labs, and $6.2 to $8.1 million for large labs. Consumables are the largest share of total annual costs in large-scale labs, whereas labor is the largest share of costs in small-scale labs. In medium-scale labs, consumables and labor have about equal shares of annual costs.

Different-sized labs differ in their capacity and efficiency. Large-scale labs test about four times the amount of cannabis per hour than medium labs, and more than 10 times what small labs test. The cost advantage of large testing labs comes from a more efficient use of inputs such as lab space, equipment, and labor. Table 6 summarizes the average of estimated testing capacities, annual costs, and testing cost per sample for each of the three lab size categories.

Cost of collection, handling and transport also vary by lab size. As of April 2018, the longest distance between a lab and a distributor in California was about 156 miles. Fig 3 shows the cost of collection, handling, and transportation (which we call the "sampling cost") per sample for distances between labs and distributors, of less than 156 miles. As expected, the longer the

**Table 6. Testing capacity, itemized annual costs, and testing cost per sample by laboratory scale.**

| Scale category | Large (n = 5) | Medium (n = 10) | Small (n = 34) |
|---|---|---|---|
| Mean number of effective samples analyzed year | 23,312 | 5,895 | 2,173 |
| Annual costs | (*Thousands*) | | |
| Capital investment, interest plus depreciation[1] | $562.02 | $378.87 | $235.38 |
| Operating and maintenance costs | | | |
| Equipment maintenance and acquisition and maintenance of ISO/IEC-17025 | $615.76 | $422.16 | $233.07 |
| Rent and basic utilities | $484.95 | $332.71 | $228.41 |
| Sales, general and administrative costs | $129.96 | $88.01 | $49.99 |
| License fees | $90.00 | $45.00 | $20.00 |
| Labor | $1,721.89 | $895.77 | $518.68 |
| Consumable costs | $3,430.88 | $866.90 | $319.35 |
| Return to risk and profit (15%) | $84.30 | $56.83 | $35.31 |
| Total costs | $7,119.76 | $3,086.25 | $1,640.18 |
| | Costs per sample | | |
| Average cost per sample of within lab testing | $306 | $525 | $757 |
| Cost of sampling (collection, transport and handling) | $8 | $13 | $21 |
| Average cost per sample of testing | $313 | $537 | $778 |

Values represent averages from 1,000 Monte Carlo simulations using distributions presented in Tables 3 and 4. Totals may not reflect sums of rows due to rounding.

distance, the higher the sampling cost. Large labs have relatively low sampling costs even at long distances. The highest possible sampling cost we assume for small labs is about $35 per sample if the distributor is located 156 miles away (Fig 3). On average, costs of collection, handling, and transportation represents a small share (about 2.5%) of total lab costs per sample.

Fig 4 shows the distribution of full testing cost per sample from 1,000 Monte Carlo simulations assuming 49 labs. Variability of the cost per sample within small labs is high, with the highest and lowest cost within that group differing by $463. The difference between the highest and lowest costs in large labs is $88, with a lowest cost per sample of about $273. The average full cost per sample tested is about $313 for large labs, $537 for medium labs, and about $778 for small labs (see Table 6).

Large cost differences per test and per batch document the large-scale economies and differences in operational efficiencies across labs of difference sizes. The aggregate amount of cannabis flowing through licensed labs in 2019 remains relatively small relative to the anticipated amounts expected in the future. That means labs that may anticipate growth, operate well below capacity. Substantial scale economies suggest that, as the market settles, the smallest labs must either expand to use their capital investment more fully, leave the industry, or provide some specialized services to distributors that are not accounted for in the analysis presented here. Simply put, the average cost differences shown in Table 6 or the simulated ranges displayed in Fig 4 should not be understood as a long run equilibrium in the cannabis testing laboratory industry.

## Testing cost per pound of cannabis marketed

Based on the shares developed based on current information, a few large labs are likely to supply almost half the testing services for cannabis sold through licensed retailers in California, while medium labs will test about 24% of cannabis and small labs about 30% of cannabis. Using these shares and the cost information documented, the weighted average of testing cost from our simulations is about $504 per sample.

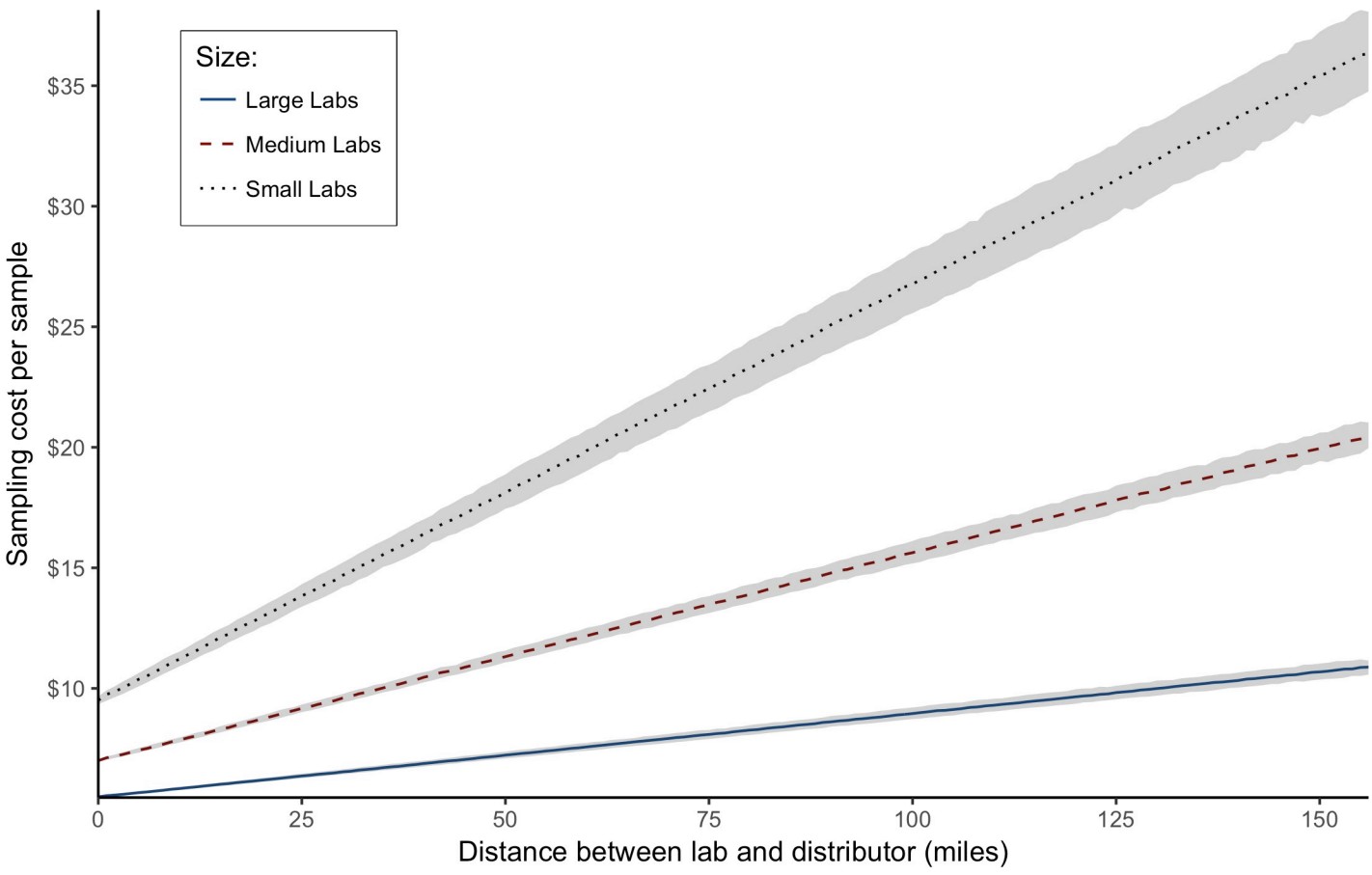

**Fig 3. Sampling cost (collection, handling, and transportation) at different distances between a lab and a distributor.**

In 2018, the first year of mandatory testing enforcement, according to official data published by the California Bureau of Cannabis Control and posted publicly on its website, failure rates in California averaged about 5.6% (not including failures due to "label claims," i.e. incorrect cannabinoid content reported on the label, which can be corrected at little cost by re-labeling without remediation). Failure rates for the first seven months of 2019, the second year of the testing regime, have averaged 4.1% [17]. We assume a 4% failure rate for the current market in California.

By comparison, in Washington State, in 2017, the second year after the testing began, 8% of the total samples failed one or more tests [19]. The Colorado Marijuana Enforcement Division reported that during the first six months of 2018, 8.9% of batches of adult-use cannabis failed testing, with infused edibles and microbial tests for flower accounting for the most failures [8].

Batch size significantly affects the per-pound testing cost of cannabis marketed, especially when batch size is smaller than 10 pounds. Fig 5 shows the costs of one pound of cannabis marketed coming from different sizes of batch flowers using 0%, 4%, and 8% rejection rates. As rejection rates increase, the differences between the costs per pound of testing different batch sizes decreases. For example, given a 0% rejection rate, the cost of testing per pound of cannabis marketed from a one-pound batch is about 27 times higher than the cost of a 48-pound batch; on the other hand, given an 8% rejection rate, the cost of testing per pound of cannabis marketed from a one-pound batch size is only seven times higher than the cost from a 48-pound batch size.

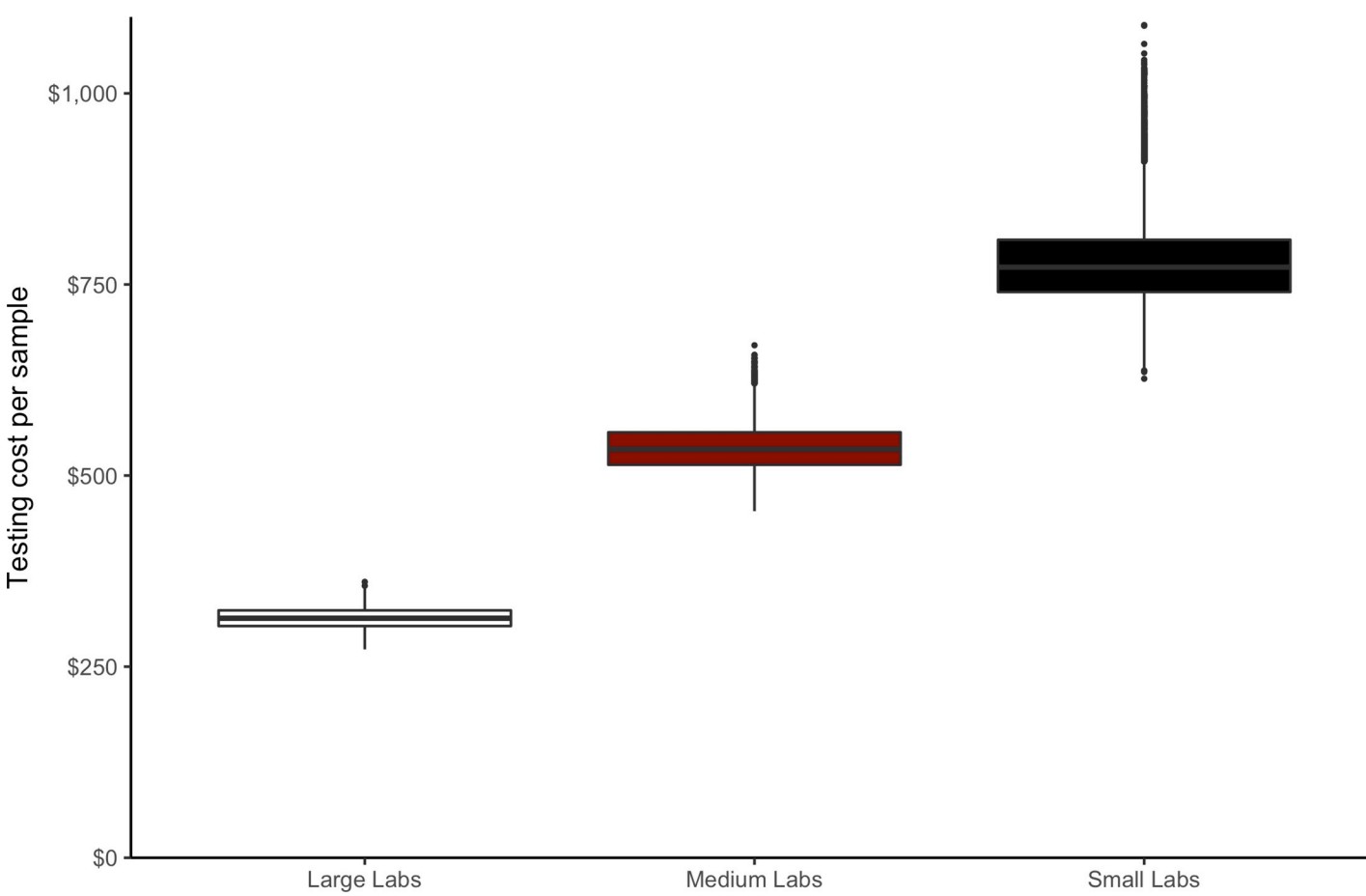

**Fig 4. Testing cost per sample from a batch of cannabis flower ($C_i$) estimated for small, medium, and large labs using 1,000 Monte Carlo simulations.**

Table 7 shows costs per pound of cannabis testing itemized into laboratory cost, the value of lost inventory, and the cost of remediating failed batches, given different rejection rates and batch sizes. For small batch sizes, laboratory costs are a higher share of total testing costs than they are for large batch sizes. For a one-pound batch size, the total cost of testing of a pound of cannabis that reaches the market is about $641 when the expected rejection rate is equal to zero. The cost increases to $714 if the expected rejection rate is equivalent to 4%, and to $791 if the expected rejection rate is 8% (Table 7).

The share of laboratory cost from total cost decreases as the rejection rate increases and the value of lost inventory therefore increases. Under an 8% expected rejection rate, the share of lost inventory is half of the total cost for eight-pound batches (Table 7).

## Discussion

In this paper, we use a simulation model to estimate the costs per pound of mandatory cannabis testing in California. To do this, we make assumptions about the cost structure and estimated the testing capabilities of labs in three different size categories, based on information collected from market participants across the supply chain. For each lab, we estimate testing cost per sample and its share, based on testing capacity, of California's overall testing supply. We then estimate a weighted average of the cost per sample and translate that value into the cost per pound of cannabis that reaches the market.

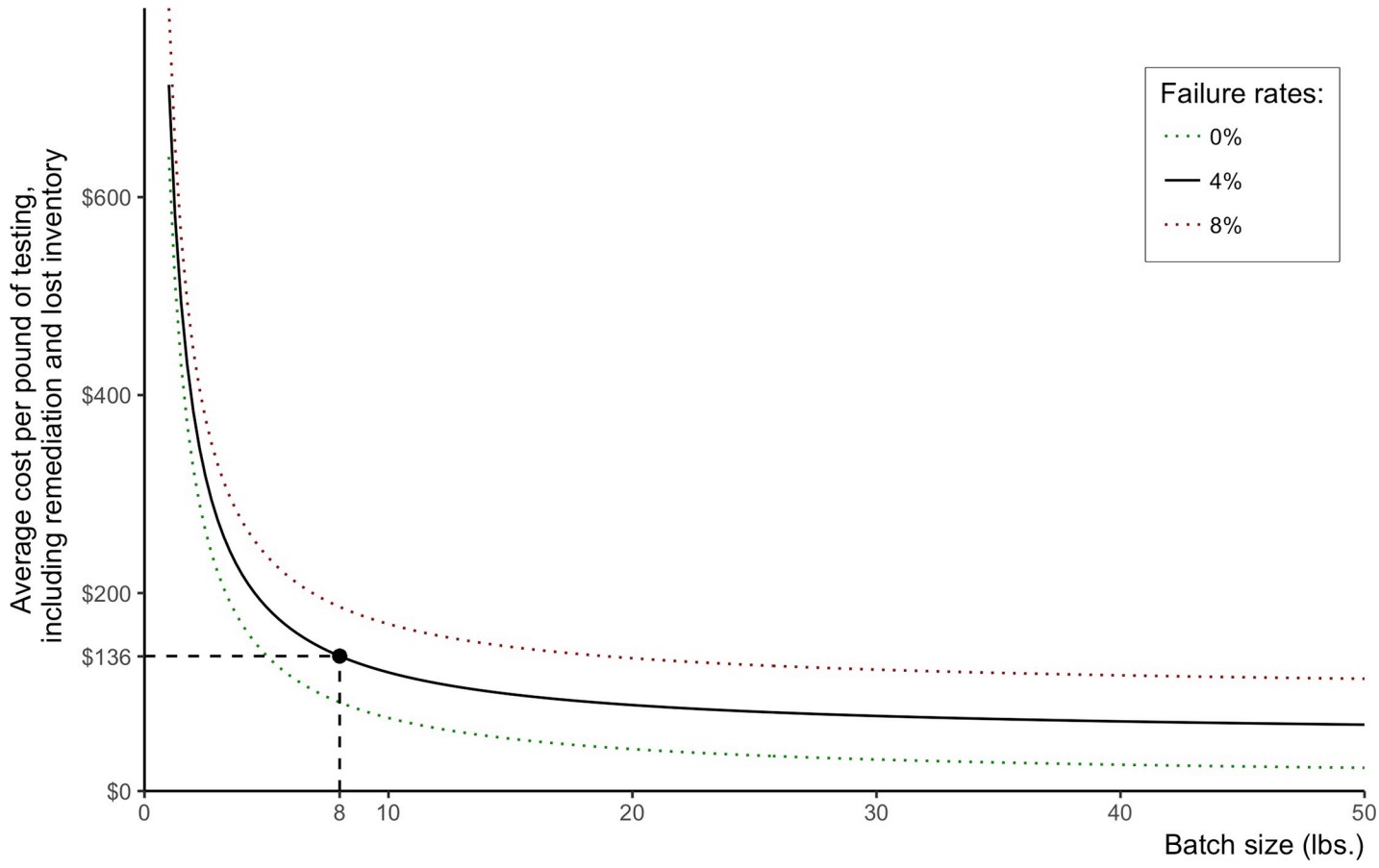

**Fig 5. Testing costs per pound of cannabis marketed at different batch sizes and rejection rates.**

We use data-based assumptions about expected rejection rates in the first and second round of testing, pre-testing, and the remediation or processing of samples that fail testing. Our simulations rely on information collected from several sources, including direct information from testing labs in California, price quotes from companies that supply testing equipment, interviews with cannabis testing experts, data on testing outcomes for cannabis and other agricultural products from California and other states, data on pesticide detection in California crops, and data on average wholesale cannabis batch sizes.

Costs needed to start a testing lab that meets California regulations depend on the scale of the lab. As lab scale rises, testing capacity rises faster than do input costs, so average costs fall with scale. We find that a large lab has four times the total costs of a small lab but 10 times the testing capacity, in part because large labs are able to use their resources (equipment, lab space, labor) more efficiently.

Testing cost per pound of cannabis marketed is particularly sensitive to batch size, especially for batch sizes under 10 pounds. Testing labs report that batch size varies widely. The maximum batch size allowed in California is 50 pounds, but many batches are smaller than 15 pounds. We assume an eight-pound average batch size in the 2019 California market, but we expect that the average batch size will increase in the future as cultivators become larger and more efficient and take advantage of the opportunity to save on testing costs (and other costs that vary nonlinearly with batch size).

**Table 7. Itemized costs per pound of cannabis that reaches the market at three rejection rates and different batch sizes.**

| Rejection Rate | Batch Size | Laboratory Cost | Value of lost inventory | Remediation Cost | Per pound cost of cannabis marketed |
|---|---|---|---|---|---|
| 0% | 1 | $634.71 | $5.95 | $0.00 | $640.66 |
| 0% | 2 | $319.81 | $5.95 | $0.00 | $325.76 |
| 0% | 8 | $83.63 | $5.95 | $0.00 | $89.58 |
| 0% | 24 | $31.15 | $5.95 | $0.00 | $37.10 |
| 0% | 48 | $18.03 | $5.95 | $0.00 | $23.98 |
| 2% | 1 | $649.53 | $26.80 | $0.20 | $676.53 |
| 2% | 2 | $327.28 | $26.80 | $0.20 | $354.28 |
| 2% | 8 | $85.58 | $26.80 | $0.20 | $112.59 |
| 2% | 24 | $31.88 | $26.80 | $0.20 | $58.88 |
| 2% | 48 | $18.45 | $26.80 | $0.20 | $45.45 |
| 4% | 1 | $664.81 | $48.29 | $0.41 | $713.52 |
| 4% | 2 | $334.97 | $48.29 | $0.41 | $383.68 |
| 4% | 8 | $87.60 | $48.29 | $0.41 | $136.30 |
| 4% | 24 | $32.62 | $48.29 | $0.41 | $81.33 |
| 4% | 48 | $18.88 | $48.29 | $0.41 | $67.59 |
| 6% | 1 | $680.57 | $70.46 | $0.63 | $751.66 |
| 6% | 2 | $342.92 | $70.46 | $0.63 | $414.01 |
| 6% | 8 | $89.67 | $70.46 | $0.63 | $160.77 |
| 6% | 24 | $33.40 | $70.46 | $0.63 | $104.49 |
| 6% | 48 | $19.33 | $70.46 | $0.63 | $90.42 |
| 8% | 1 | $696.83 | $93.34 | $0.85 | $791.02 |
| 8% | 2 | $351.11 | $93.34 | $0.85 | $445.30 |
| 8% | 8 | $91.82 | $93.34 | $0.85 | $186.01 |
| 8% | 24 | $34.20 | $93.34 | $0.85 | $128.39 |
| 8% | 48 | $19.79 | $93.34 | $0.85 | $113.98 |

Testing itself is costly, but losses inflicted by destroying cannabis that fails testing is a major component of overall costs. Low or zero tolerance levels for pesticide residues are the most demanding requirement, and result in the greatest share of safety compliance testing failures [17]. Cannabis standards are very tight compared to those for food products in California. A significant share of tested samples from California crops have pesticide residues that would be over the tolerance levels established for California cannabis (see Table 5). Some foods that meet pesticide tolerance established by California EPA may be combined with dried cannabis flowers to generate processed cannabis products (such as baked goods, for example). Pesticide residues coming from the food inputs may generate detection levels of pesticide over the tolerance levels set by cannabis law and regulation, even if they are otherwise compliant as food products. Cannabis testing regulation is strict compared to tobacco, another inhalable crop. Tobacco has no pesticide tolerance limits because it is considered to be an inedible crop used for recreational purposes [5]. Cannabis has multiple pathways of intake, such as edibles, inhalable, patches, etc., and also may be prescribed for people with a health condition, searching for alternatives to traditional medicine.

Some labs report that when samples barely fail one test, they have a policy of re-testing that sample to reduce the probability of false positives. Some labs have reported up to 10% in variation in test results from the same sample. Some labs indicate that about 25% of samples need to be re-tested to be sure that results are accurate. Such concerns have been widely reported. In July 2018, some producers voluntarily recalled cannabis products after receiving inconsistent

results of contaminant residues from different laboratories [20]; and some California labs have also been sanctioned by the Bureau of Cannabis Control for failing state audits on pesticide residue tests.

A major issue for legal, taxed and licensed cannabis market is competition with cannabis marketed through untaxed and unlicensed segment. Higher testing costs translate into higher prices in the licensed segment. Safety regulations and testing may improve the perceived safety and quality of cannabis in the licensed segment, thus adding value for some consumers [21]. However, price-sensitive consumers move to the unlicensed segment when licensed cannabis gets too expensive. A useful avenue for further research is to investigate cannabis testing regulations and standards across states to assess implications for consumer and community wellbeing and competition with unlicensed cannabis.

Compared with other agricultural and food industries, the licensed cannabis industry in California has relatively little data. Banking is still done in cash, and sources of government financial data are less available for cannabis than they are for other industries. As the licensed cannabis segment develops, we expect that increased access to data on the market for testing services, including on prices, quantities, and batch sizes. Data from tax authorities, the track-and-trace system, and the licensing system will then help clarify the costs and implications of mandatory cannabis testing.

## Supporting information

**S1 Appendix.**
(DOCX)

## Author Contributions

**Conceptualization:** Pablo Valdes-Donoso, Daniel A. Sumner.

**Data curation:** Pablo Valdes-Donoso, Robin Goldstein.

**Formal analysis:** Pablo Valdes-Donoso, Daniel A. Sumner.

**Funding acquisition:** Daniel A. Sumner.

**Investigation:** Pablo Valdes-Donoso.

**Methodology:** Pablo Valdes-Donoso.

**Project administration:** Daniel A. Sumner.

**Supervision:** Daniel A. Sumner.

**Validation:** Pablo Valdes-Donoso, Daniel A. Sumner.

**Visualization:** Pablo Valdes-Donoso, Robin Goldstein.

**Writing – original draft:** Pablo Valdes-Donoso.

**Writing – review & editing:** Daniel A. Sumner, Robin Goldstein.

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
