## [Decision Letter · Decision Letter 0]

5 Mar 2020

PONE-D-19-35034

Costs of cannabis testing compliance: Assessing mandatory testing in the California cannabis market

PLOS ONE

Dear Dr Valdes-Donoso,

Thank you for submitting your manuscript to PLOS ONE. After careful consideration, we feel that it has merit but does not fully meet PLOS ONE’s publication criteria as it currently stands. Therefore, we invite you to submit a revised version of the manuscript that addresses the points raised during the review process.

We would appreciate receiving your revised manuscript by Apr 12 2020 11:59PM. To enhance the reproducibility of your results, we recommend that if applicable you deposit your laboratory protocols in protocols.io, where a protocol can be assigned its own identifier (DOI) such that it can be cited independently in the future. For instructions see: http://journals.plos.org/plosone/s/submission-guidelines#loc-laboratory-protocols

We look forward to receiving your revised manuscript.

Kind regards,

Renuka Sane

Academic Editor

PLOS ONE

Additional Editor Comments (if provided):

This is a very well done study, as also reflected in the referee's reports. Please address some of the questions raised by the referee for final publication.

Journal Requirements:

2. We note that you have stated that you collected financial, managerial, and logistics data from managers or representatives of currently operating or prospective licensed testing labs. Please ensure that you have outlined the steps taken to ensure the data collected from these labs were suitably representative. Please include details of how the labs were selected and how many refused to provide their data or did not respond. Please also see our policy for citation of personal communications here: https://journals.plos.org/plosone/s/submission-guidelines#loc-references.

3. We note that Figure 2 in your submission contains map images which may be copyrighted. All PLOS content is published under the Creative Commons Attribution License (CC BY 4.0), which means that the manuscript, images, and Supporting Information files will be freely available online, and any third party is permitted to access, download, copy, distribute, and use these materials in any way, even commercially, with proper attribution. For these reasons, we cannot publish previously copyrighted maps or satellite images created using proprietary data, such as Google software (Google Maps, Street View, and Earth). For more information, see our copyright guidelines: http://journals.plos.org/plosone/s/licenses-and-copyright.

You may seek permission from the original copyright holder of Figure 2 to publish the content specifically under the CC BY 4.0 license. 

If you are unable to obtain permission from the original copyright holder to publish these figures under the CC BY 4.0 license or if the copyright holder’s requirements are incompatible with the CC BY 4.0 license, please either i) remove the figure or ii) supply a replacement figure that complies with the CC BY 4.0 license. Please check copyright information on all replacement figures and update the figure caption with source information. If applicable, please specify in the figure caption text when a figure is similar but not identical to the original image and is therefore for illustrative purposes only.

4. Please ensure that you refer to Figure 5 in your text as, if accepted, production will need this reference to link the reader to the figure.

Reviewers' comments:

Reviewer's Responses to Questions

**Comments to the Author**

1. Is the manuscript technically sound, and do the data support the conclusions?

Reviewer #1: Yes

2. Has the statistical analysis been performed appropriately and rigorously? 

Reviewer #1: Yes

3. Have the authors made all data underlying the findings in their manuscript fully available?

Reviewer #1: Yes

4. Is the manuscript presented in an intelligible fashion and written in standard English?

Reviewer #1: Yes

5. Review Comments to the Author

Reviewer #1: This is a wonderfully detailed study.

The idea to model sampling costs as a function of failure rates is direct and intuitive. Of course, tightening rules by lowering tolerance thresholds or adding prohibited substances will generally raise failure rates, and loosening rules would lower them.

The authors 1) model sample testing costs as a function of failure rates; 2) accounting for economies of scales, model costs for small, medium, and large laboratories; 3) assuming market shares of 70%, 20%, and 10%, for large, medium, and small laboratories, simulate average costs per sample, among many other model outputs.

At each step, models are very detailed, with generous information provided by the authors’ survey. For instance, laboratory economies of scale are generated considering variable failure rates, operational efficiency, and operating days per year. Ultimately large labs are modeled to have four times the cost of small labs but ten times the capacity, suggesting potential for the industry to consolidate.

This provides a general framework for the evaluation of cannabis testing costs, and I hope the authors’ model may be used as a basis for further research modeling the price effects of cannabis testing regulations.

It is interesting to wonder how this measure fares across states. What exactly would you be measuring: the competency of the laboratories, the stringency of the regulations, or the quality of the local product?

Minor notes:

“Costs needed to start a testing lab that meets California regulations depend [ON] the scale of the lab “ (p31). Insert “ON”.

The authors compare testing standards for cannabis and food products. However, they do not seem to approach the question: is this an equal comparison? Does smoking or vaporizing a substance put the user at greater exposure to health risks than eating it? If somehow heating or combustion or exposure to the mouth, throat, and lungs puts a user at greater health risk, that may justify the more severe standards applied to cannabis.

The authors may consider also comparing adding tobacco standards to their comparison of cannabis and food products (or remarking briefly on the comparison), to provide a comparison of regulations pertaining to another combustible intended for inhalation.

6. PLOS authors have the option to publish the peer review history of their article (what does this mean?). If published, this will include your full peer review and any attached files.

Reviewer #1: Yes: Steven Davenport

---

## [Author Response · Author response to Decision Letter 0]

19 Mar 2020

Dear Editor,

Thank you for offering us the opportunity to revise our manuscript for PLOS ONE. We have carefully revised the manuscript to address the comments of you and the reviewer and meet editorial guidelines.

Please see our responses below:

We have revisited the PLOS ONE style templates and made the following changes:

1. We have replaced the abbreviation of the state of California in authors’ affiliation.

We have included an indent at the start of a new section.

2. We now cite figures as Fig 

2. We note that you have stated that you collected financial, managerial, and logistics data from managers or representatives of currently operating or prospective licensed testing labs. Please ensure that you have outlined the steps taken to ensure the data collected from these labs were suitably representative. Please include details of how the labs were selected and how many refused to provide their data or did not respond. Please also see our policy for citation of personal communications here: https://journals.plos.org/plosone/s/submission-guidelines#loc-references.

We now have outlined the steps taken to collect data from representative labs and included the set of questions to which they responded. We have included a supplementary document (S1 Appendix) with the set of questions used as a guideline for the interviews. Within S1 Appendix, we describe the confidentially agreements discussed with interviewees in such a way that their names and details business information would not be disclosed. Please see Lines 230-241 of the document Revised Manuscript with Track Changes.

3. We note that Figure 2 in your submission contains map images which may be copyrighted. All PLOS content is published under the Creative Commons Attribution License (CC BY 4.0), which means that the manuscript, images, and Supporting Information files will be freely available online, and any third party is permitted to access, download, copy, distribute, and use these materials in any way, even commercially, with proper attribution. For these reasons, we cannot publish previously copyrighted maps or satellite images created using proprietary data, such as Google software (Google Maps, Street View, and Earth). For more information, see our copyright guidelines: http://journals.plos.org/plosone/s/licenses-and-copyright.

1. You may seek permission from the original copyright holder of Figure 2 to publish the content specifically under the CC BY 4.0 license. 

Figure 2 contains a map that we created using R software. R is free software, so as the packages that are needed to create figures and maps (e.g., ggplot, maps, etc.). We included a sentence in Lines 358-359 that explicitly indicates that the map has been created using R software and the zip codes published by the BCC. We have included the cites of these sources [15 and 16]. 

4. Please ensure that you refer to Figure 5 in your text as, if accepted, production will need this reference to link the reader to the figure.

We have included the reference and title of Figure 5. Please see Line 553 and Lines 564-565 with our edits. 

Reviewers' comments:

Reviewer's Responses to Questions

Comments to the Author

1. Is the manuscript technically sound, and do the data support the conclusions?

Reviewer #1: Yes

2. Has the statistical analysis been performed appropriately and rigorously? 

Reviewer #1: Yes

3. Have the authors made all data underlying the findings in their manuscript fully available?

Reviewer #1: Yes

4. Is the manuscript presented in an intelligible fashion and written in standard English?

Reviewer #1: Yes

5. Review Comments to the Author

Reviewer #1: This is a wonderfully detailed study.

The idea to model sampling costs as a function of failure rates is direct and intuitive. Of course, tightening rules by lowering tolerance thresholds or adding prohibited substances will generally raise failure rates, and loosening rules would lower them.

The authors 1) model sample testing costs as a function of failure rates; 2) accounting for economies of scales, model costs for small, medium, and large laboratories; 3) assuming market shares of 70%, 20%, and 10%, for large, medium, and small laboratories, simulate average costs per sample, among many other model outputs.

At each step, models are very detailed, with generous information provided by the authors’ survey. For instance, laboratory economies of scale are generated considering variable failure rates, operational efficiency, and operating days per year. Ultimately large labs are modeled to have four times the cost of small labs but ten times the capacity, suggesting potential for the industry to consolidate.

This provides a general framework for the evaluation of cannabis testing costs, and I hope the authors’ model may be used as a basis for further research modeling the price effects of cannabis testing regulations.

It is interesting to wonder how this measure fares across states. What exactly would you be measuring: the competency of the laboratories, the stringency of the regulations, or the quality of the local product?

We agree that this is an interesting question and should be subject to further research. In this article however, we focused on detailed analysis of data from California. We describe the cannabis testing regulations in other states that allow recreational cannabis in the “Background” subsection. We have added a sentence about cross-state comparisons in the brief discussion of further research in the concluding section.

Minor notes:

“Costs needed to start a testing lab that meets California regulations depend [ON] the scale of the lab “ (p31). Insert “ON”.

Thank you. We have included [ON] in Line 601of the document Revised Manuscript with Track Changes

The authors compare testing standards for cannabis and food products. However, they do not seem to approach the question: is this an equal comparison? Does smoking or vaporizing a substance put the user at greater exposure to health risks than eating it? If somehow heating or combustion or exposure to the mouth, throat, and lungs puts a user at greater health risk, that may justify the more severe standards applied to cannabis.

There may be higher risk of health impacts if a crop is contaminated with microbial or fungus and consumed via inhalers, we do not know if there is a different risk by consuming a product contaminated with pesticides. Cannabis and cannabis products are offered to be consumed via edibles or inhalable. Here we treat cannabis like any other crop in California, thus we do such a comparison. We have included the sentence in Lines 427-429 to make this explicit.

The authors may consider also comparing adding tobacco standards to their comparison of cannabis and food products (or remarking briefly on the comparison), to provide a comparison of regulations pertaining to another combustible intended for inhalation.

We did not elaborate on tobacco standards because tobacco is not a relevant crop for California (California accounts for an extremely low share of U.S. tobacco production. Of course, tobacco continues to be consumed and we have included a brief comparison between tobacco standards and cannabis (lines 627-631). 

6. PLOS authors have the option to publish the peer review history of their article (what does this mean?). If published, this will include your full peer review and any attached files.

Do you want your identity to be public for this peer review? For information about this choice, including consent withdrawal, please see our Privacy Policy.

Reviewer #1: Yes: Steven Davenport

---

## [Editor Report · Decision Letter 1]

7 Apr 2020

Costs of cannabis testing compliance: Assessing mandatory testing in the California cannabis market

PONE-D-19-35034R1

Dear Dr. Valdes-Donoso,

We are pleased to inform you that your manuscript has been judged scientifically suitable for publication and will be formally accepted for publication once it complies with all outstanding technical requirements.

With kind regards,

Renuka Sane

Academic Editor

PLOS ONE
---

## [Editor Report · Acceptance letter]

9 Apr 2020

PONE-D-19-35034R1 

Costs of cannabis testing compliance: Assessing mandatory testing in the California cannabis market 

Dear Dr. Valdes-Donoso:

I am pleased to inform you that your manuscript has been deemed suitable for publication in PLOS ONE. Congratulations! Your manuscript is now with our production department. 

With kind regards,

on behalf of

Dr. Renuka Sane 

Academic Editor

PLOS ONE